# Experimental Investigations of Distributed Fiber Optic Sensors for Water Pipeline Monitoring

**DOI:** 10.3390/s23136205

**Published:** 2023-07-06

**Authors:** Manuel Bertulessi, Daniele Fabrizio Bignami, Ilaria Boschini, Marina Longoni, Giovanni Menduni, Jacopo Morosi

**Affiliations:** 1Civil and Environmental Engineering Department, Politecnico di Milano, Piazza Leonardo da Vinci, 32, 20133 Milano, Italy; ilaria.boschini@polimi.it (I.B.); marina.longoni@mail.polimi.it (M.L.); giovanni.menduni@polimi.it (G.M.); 2Fondazione Politecnico di Milano, Piazza Leonardo da Vinci, 32, 20133 Milano, Italy; daniele.bignami@fondazione.polimi.it; 3Cohaerentia S.r.l, Via Pinturicchio 5, 20131 Milano, Italy; jacopo.morosi@cohaerentia.com

**Keywords:** pipeline health monitoring, Brillouin DFOS, water leakage detection, smart pipe

## Abstract

Water Loss (WL) is a global issue. In Italy, for instance, WL reached 36.2% of the total fresh water conveyed in 2020. The maintenance of a water supply system is a strategic task that requires a huge amount of investment every year. In this work, we focused on the use of Distributed Fiber Optic Sensors (DFOS) based on Stimulated Brillouin Scattering (SBS) technology for monitoring water pipeline networks. We worked on High-Density Polyethylene (HDPE) pipes, today the most widely used for creating water pipelines. By winding and fixing the optic fiber cable on the pipe’s external surface, we verified the ability to detect strain related to pressure anomalies along a pipeline, e.g., those caused by water leakage. We performed two experimental phases. In the first one, we assessed the sensibility of sensor layout on an HDPE pipeline solicited with static pressure. We investigated the viscoelastic rheology of the material by calibrating and validating the parameters of a Burger model, in which Maxwell and Kelvin-Voigt models are connected in series. In the second experimental phase, instead, we focused on the detection of the pressure anomaly produced by leakage in a pipeline circuit set up with running water moved by a pump. The theoretical and experimental studies performed returned overall positive feedback on the use of DFOS for the monitoring of HDPE water pipelines. Future developments will be focused on more detailed studies of this monitoring solution and on the industrial production of “natively smart” HDPE pipes in which DFOS cables are integrated into the pipeline surface during the extrusion process.

## 1. Introduction

Water is notoriously a strategic and fundamental resource for the social welfare and economic growth of any nation. An efficient water resources management policy is nowadays a common goal among governments all over the world to prevent water wastage. Moreover, the availability of this primary good is increasingly threatened by climate change’s impact on the hydrological chain [1].

With regard to water supply networks, Water Loss (WL) is a global issue: 48.6 billion cubic meters are lost per year in the world [2]. In Italy, for instance, WL reached 36.2% of the total water conveyed in 2020, equal to 0.9 billion cubic meters (National Institute of Statistics ISTAT, 2022). WL affects the technical stability of the water supply system and its operational age, as well as the quantity and quality of the water transported [3]. Water pipeline failure mechanisms can be categorized into three groups: pipe-intrinsic, operational, and environmental [4]. The first category includes age, chemical degradation, and manufacturing defects in the pipe and/or in the joint system. The second pipe failure category deals mainly with network operations that can produce cyclical pressure and transient pressure surge. This effect, also known as “water hammer”, overpressures the pipe and represents an important cause of failure, especially in large diameters pipes [5]. Finally, the last category embraces all pipe failures caused by sudden or huge seasonal changes; thermal ones are the most related to failures [6]. 

Given this framework, the maintenance of a water supply system is a strategic task that requires a huge amount of investment every year. A pipeline failure, depending on the type of fluid and the discharge transported, can produce heavy damage to the surrounding exposure. Consider, for example, the flood risk induced downstream by a hidden failure of a penstock conveying water to a hydro-power plant [7]. This can be mitigated by installing a monitoring system able to detect in time penstock deformation trends not compliant with the operational state. 

Concerning pipeline monitoring systems, there are several solutions available in the literature, most of them related to the oil and gas supply field.

Hardware-based monitoring techniques can be divided into two main categories: non-invasive when the sensors are installed outside the pipe and invasive when they are placed inside it [8]. The first category includes Visual Inspection (VI) for aboveground pipelines and Ground Penetrating Radar (GPR) for underground networks. However, this technology is inefficient in detecting water leaks in non-metallic pipes and is today commonly used for water supply networks [9]. Invasive techniques require instead the installation of sensors internally in the pipe to detect pressure, flow rate, temperature, water density, and viscosity. Acoustic sensors, for instance, can detect small vibrations produced by the water rapping out from the pipeline [10]. The installation procedure of these sensors must be very precise and accurate in order to avoid the creation of sources of disturbance to the regular flow of the fluid conveyed. Moreover, also in the case of Wireless Sensors Networks (WSN), they need an energy supply, and this fact constitutes a costly issue for long pipeline infrastructures placed outside urban areas.

Software-based methods, instead, localize leaks by continuously monitoring and elaborating on one or more internal variables. Some of these methods perform statistical analysis on measurements, while others take into account physical laws [11]. An innovative solution in this field is represented by the use of Machine Learning (ML) algorithms on raw data to build prediction models on the presence of leaks [12]. 

In this work, we focused on the use of Fiber Optic Sensors (FOS) for monitoring water pipeline networks. The use of this type of sensor for Structural Health Monitoring (SHM) and Geotechnical Health Monitoring (GHM) is widely used for its great adaptability to each kind of infrastructure and its immunity to electromagnetic interferences [13].

Fiber Bragg Grating (FBG) sensors are quasi-distributed FOS used to detect any mechanical change to the pipeline state, such as bending, impact, and fatigue [14]. FBGs have been installed in several environments, including subsea oil and gas pipelines [15]. These sensors can detect the Negative Pressure Wave (NPW) produced by the leakage [16]. Indeed, when a leak occurs, a pressure drop wave propagates from the leakage point toward both ends of the pipeline changing the hoop strain. Hence, a series of FBG hoop strain sensors installed along the pipeline can detect with high accuracy the leakage position by the time of arrival of the wave and the entity of the hoop strain variation. However, these sensors offer punctual strain monitoring, and in the case of pipeline networks, commonly 10 to hundreds of kilometers long, they represent an expensive solution due to the important quantity of non-sensing cable required to power and connect them. 

Fully Distributed Fiber Optic Sensors (DFOS) are instead a competitive solution for the monitoring of large civil, hydraulic, and geotechnical structures and infrastructures [17,18]. With this technology, the fiber optic cable is sensing for all its length, and this feature well suits the typical linear nature of infrastructures. Moreover, the energy supply is needed only by the interrogation unit, which can also be placed a few kilometers far from the monitored structure. The increasingly competitive costs of system components open the path to extensive use of DFOS, even integrated into structural materials or revetments [19,20,21]. 

DFOS’s main applications in the Pipeline Health Monitoring (PHM) field are all related to oil and gas supply systems. Two main technologies are commonly used for this purpose: Stimulate Brillouin Scattering (SBS) and Distributed Acoustic Sensing (DAS).

SBS is the most affirmed and broadly applied solution of the DFOS family. Its success is surely given by the fact that the Brillouin effect can be used on kilometric standard monomodal telecommunication optical fibers [22]. In this work, we used the most widely used Brillouin sensing technique, the Brillouin Optical Time Domain Analysis (BOTDA). The technical details of this technology will be illustrated in Section 2. SBS-based interrogation units are increasingly cost-effective and provide an absolute measurement related to the linear superposition of mechanical strain and temperature. This means that the system has memory and can be set to a given acquisition time, reducing energy consumption. 

DAS belongs instead to the Rayleigh-based DFOS family, and it can detect local pipeline vibrations induced by pipeline leaks [23]. Gas pipeline leaks of the order of 1% have been detected in the experimental investigations of Stajanca et al. [24], showing a sensitivity much higher than Brillouin-based DFOS. However, the experiments were performed in static conditions without the vibration background of a medium flowing through. The vibrational signature of a leak is also characterized by high variability, and it strongly depends both on the pipeline geometry and its location and the entity of the leak itself. Hence, this method requires an important post-processing phase to filter parasites’ vibrational modes and detect the presence of those related to leaks. Moreover, in the literature, there is no evidence of the workability of this method on HDPE pipes that are the target of our investigations. Regarding the measurements, those coming from DAS are not absolute values, but they return a temperature or strain variation concerning the instant in which the interrogator unit is started. In this sense, a DAS system has to be continuously active to follow properly the pipeline health history, in particular when important solicitations occur. 

The installation layouts commonly tested in literature for PHM are three: axial, hoop-loop, or helical.

Axial geometry is of benefit for leaks and pipeline dislocation detection. In the first case, DFOS cable can be simply positioned in a loose layout, that is, without mechanical bonding [25]. When the fluid transported escapes from the pipeline, it changes the local soil temperature, and this variation is enough to be detected by DFOS [26]. Pipeline dislocation, instead, is perceivable by bonding three DFOS cables displaced by 120° around the circumference of the pipeline [27]. From strain measurement along these three lines and considering the Euler–Bernoulli beam theory, it is possible to detect pipeline dislocation and also estimate the direction along which it occurs.

In hoop or loop configuration, instead, the DFOS cable curves around the outer surface of the pipeline to detect hoop strain variations induced by pressure anomalies related to leakages or pipeline corrosion. In the first case, DFOS measures the hoop strain variation produced by the leakage, as it will be shown in this work. Concerning corrosion, hoop strain is strictly linked to the pipeline thickness, and this parameter can change with time and locally due to corrosion phenomena. Experiments performed on a steel pipeline corroded with a controlled process showed that thickness variations of the order of mm can be measured by DFOS cable hoops [28]. 

Finally, the helical configuration is a variation of the previous one, but in this case, the pipeline is wrapped continuously in the DFOS cable, avoiding connectors or unbonded cable portions. 

This work aims at testing this latter sensor layout on High-Density Polyethylene (HDPE) pipes, today the most widely used for creating water pipelines up to 300 mm in diameter [29]. This material offers low failure rates and good resistance to corrosive soils, flexibility to soil movements, and transient pressure surge. Moreover, HDPE better withstands huge thermal excursions concerning iron and steel pipes [30]. By winding the fiber optic cable on the pipe’s external surface, we verified the ability to detect strain related to pressure anomalies along a pipeline, e.g., those caused by water leakage. In this way, we want to test the performance of this technology applied to the water supply field as a cost-effective monitoring solution to mitigate WL. Our investigation starts with urban water pipelines by considering HDPE pipes from small to medium diameter. However, the monitoring principle can be easily extended to larger pipelines made of different materials, e.g., aboveground steel water penstocks.

We performed two experimental phases. In the first one, we assessed the sensor layout on an HDPE pipeline with static pressure. The main objective was to evaluate the strain sensitivity of this solution on the simple pipe expansion given by the internal pressure. Viscoelastic phenomena have been observed and taken into account in the creation of a digital twin of the experimental setup. This latter has been modeled to compare the experimental results with the analytical solution and verify the strain sensitivity of the sensor layout.

In the second experimental phase, instead, we focused on the detection of the pressure anomaly produced by leakage in a pipeline circuit set up with running water moved by a pump. The leakage was controlled by a gate valve to vary the outgoing discharge and find the minimum pressure anomaly perceivable by the DFOS system.

Both phases returned positive feedback on the measurement consistency and on the capacity of DFOS to monitor strain anomalies—hence pressure anomalies—in HDPE pipes. It should be stressed here that the rheological behavior of HDPE shows a strong and complex viscoelastic character quite far from the simple linear elasticity of the steel. This aspect is particularly relevant wherever it is desired to guarantee a definitively quantitative character to the measure. Further investigations are required for the detection of other phenomena, e.g., pipe dislocation, and for DFOS installation on pipes from an industrial perspective.

The article is subdivided as follows: in Section 2, the materials and the experimental setups implemented in both phases are broadly described. Also, the hydraulic theoretical background and DFOS technology adopted are broadly illustrated, as well as the raw data elaboration process to get distributed strain measurements of the pipe. Section 3 shows the main results obtained from the two experimental phases, whereas in Section 4, we analyze and discuss them. This section ends with our conclusions on these experimental investigations and the possible future developments of this research. 

## 2. Materials and Method

### 2.1. Sensor Technology and Layout

The sensing cable used in the experiments is a standard single-mode tight buffer optical fiber (SM G652D) having a diameter of 0.9 mm and a polymeric unarmoured coating that meets LSZH specifications. The cable is helically wrapped on the external surface of the pipe with uniform spacing, as shown in Figure 1. The bonding is simply provided by duct tape that follows the cable path. The sensor cable simultaneously detects both strain and temperature variations. At the end of the pipe, a portion of the sensing cable returning to the interrogator is loosely fixed to the pipe for the thermal effect compensation on DFOS measurement [31].

The DFOS interrogator—provided by Cohaerentia Srl—relies on the standard BOTDA technique, and its block diagram is reported in Figure 2. The working principle is based on the Stimulated Brillouin Scattering (SBS), which is the interaction between a pulsed pump wave and a continuous-wave (CW) counter-propagating probe [32].

The light coming from a 1530 nm Distributed Feedback (DFB) laser is divided into two branches by a 3 dB optical coupler. The top branch is sinusoidally modulated by a Mach–Zehnder Modulator (MZM) driven by an RF generator to provide a continuous-wave (CW) probe signal launched from the far end of the sensing fiber through an optical circulator. Its frequency is downshifted by approximately 10.8 GHz from the pump frequency and scanned over a 1 GHz range (equivalent to a dynamic range of 20 mε). A passive depolarizer is used to avoid polarization fading [33]. In the bottom branch, another Mach–Zehnder modulator, driven by a pulse generator and followed by an Erbium Doped Fiber Amplifier (EDFA), is used to generate 20 ns pump pulses (equivalent to a 2 m spatial resolution) with a 30 kHz repetition rate, launched into the sensing fiber from a second optical circulator. The same circulator also routs the probe signal to a 125 MHz photodetector (PD). For each position along the sensing fiber, the frequency scan of the probe light allows us to reconstruct the Brillouin Gain Spectrum (BGS) due to the backscattering signal generated by the pulsed pump. The specific frequency corresponding to the maximum of the BGS—called the Brillouin Frequency Shift (BFS)—depends on the local temperature and strain acting on the sensing fiber. After the PD, the signal is sampled by a 125 MS/s Analog-to-Digital Converter (ADC) card (equivalent to an 80 cm spatial sampling) and processed to recover the BFS profile along the whole length of the sensing cable. The measurement accuracy of the employed BOTDA interrogator has been verified to be better than 0.5 °C/10 µε.

Given the reference state $t_{0}$, the BFS at a generic instant t is the linear composition of strain and temperature variation concerning the former [34]:(1)$$\Delta {\upsilon_{BFS,\Delta t}=\upsilon }_{BFS,t}-\upsilon_{BFS,t_{o}}=C_{T}∆T+C_{\varepsilon }\varepsilon ,$$
where $C_{T}$ [MHz/°C] and $C_{\varepsilon }$ [MHz/µε] are the thermo-optical and strain sensitivity coefficients which, for a standard fiber-optic cable such as that used here, correspond to 1 MHz/°C and 0.05 MHz/µε, respectively [35]. However, these parameters are highly influenced by the cable coating properties and the bonding conditions, which can modify their value [36]. In this work, we assume the standard values of the coefficients. By considering the BFS with respect to a reference state, it is possible to eliminate the pre-strain component on the DFOS cable caused by the installation procedure. In each experimental case, the reference state has been taken before the perturbing cause, i.e., before pumping water in phase I and creating the water leakage in phase II. 

For each sampling point, strain and temperature variations are integrated on a spatial interval of 2 m, which defines the effective spatial resolution of the interrogator. Considering the sensor layout, the effective linear spatial resolution of the system along the pipe depends on the external pipe diameter and the spacing chosen in mounting the cable coil.
(2)$$\Delta s=\frac{i}{L_{c}}S,$$
where i is the interrogator spatial resolution [m], S [m] is the coil spacing, and $L_{c}$ [m] is the coil length equal to
(3)$$L_{c}=\sqrt{S^{2}+{C_{e}}^{2}},$$
where $C_{e}$ [m] is the pipe’s external circumference.

Table 1 reassumes the value of these parameters for each experimental phase performed.

The total measurement time ∆t has been tuned according to the experimental needs by properly setting the frequency scanning window Δν, the frequency step $\nu_{s}$ and the number of averages $N_{avg}$ per frequency step, as reported in Table 1. In phase I, ∆t has been set to 7 s to strictly follow the viscoelastic behavior of the pipe as a pressure step was applied. In phase II, the focus was on the static effects caused by the leakage on the deformational regime; hence ∆t was set to 20 s to further increase the measurement accuracy. 

### 2.2. Phase I: Assessment of the Sensitivity of the Sensor Layout

Phase I aimed at characterizing the sensitivity of the chosen layout, i.e., the ability of the DFOS cable, helically wrapped and fixed on the pipe surface using duct tape, to properly detect circumferential or hoop strain $\varepsilon_{\theta }$. This strain is theoretically given by:(4)$$\varepsilon_{\theta }=\frac{u}{r_{e}},$$
where u is the radial displacement and $r_{e}$ is the pipe’s outer radius. The radial displacement depends on the geometry of the pipe section, on the mechanical properties of the material composing the pipe (Young’s modulus E and Poisson’s coefficient ν) and on the internal pressure $p_{i}$ [37]:(5)$$u=(2-\nu )\frac{\beta^{2}}{1-\beta^{2}}\frac{r_{e}p_{i}}{E},$$
where β [−] is the ratio between the internal and the external radius of the pipe. Considering Equations (4) and (5), we have:(6)$$\varepsilon_{\theta }=(2-\nu )\frac{\beta^{2}}{1-\beta^{2}}\frac{p_{i}}{E},$$

As for all thermoplastic materials, HDPE’s response to the instantaneous stress caused by the internal pressure $p_{i}$ is not simply governed by Hooke’s law due to its viscoelastic rheology [38]. $\varepsilon_{\theta }$ is the combination of an instantaneous elastic component $\varepsilon_{\theta ,0}$ and a retarded time-dependent viscous or creep component $\varepsilon_{\theta ,r}(t)$ [39]:(7)$$\varepsilon_{\theta }\left(t\right)=\varepsilon_{\theta ,0}+\varepsilon_{\theta ,r}(t),$$

This mechanical behavior is well reproduced by a Burger model, in which the Maxwell and Kelvin–Voigt models are connected in series (Figure 3). This model is broadly used for modeling the first and the second characteristic stage of creeping in polymeric materials [40]. The model is governed by the following law:(8)$$\varepsilon_{\theta }(t)=\sigma_{0}\left(\frac{1}{E_{1}}+\frac{1}{E_{2}}\left(1-e^{-\frac{E_{2}t}{{ⴄ}_{2}}}\right)+\frac{t}{{ⴄ}_{3}}\right),$$
where $\sigma_{0}$ is instantaneous constant stress, $E_{1}$ [Pa] is the elastic modulus, $E_{2}$ [Pa] is the modulus of elasticity of the creep response and ${ⴄ}_{2}$ and ${ⴄ}_{3}$ [Pa s] are the coefficients of dynamic viscosity. The ratio ${ⴄ}_{2}/E_{2}$ determines the retardation or response time τ [s] of the material.

By substituting Equation (8) in Equation (6) and also the thermal deformation component in case of a temperature variation Δ*T* [°C], the final mechanical model of the HDPE pipe is given by:(9)$$\varepsilon_{\theta }\left(t\right)=\left(2-\nu \right)\frac{\beta^{2}}{1-\beta^{2}}p_{i}\left(\frac{1}{E_{1}}+\frac{1}{E_{2}}\left(1-e^{-\frac{E_{2}t}{{ⴄ}_{2}}}\right)+\frac{t}{{ⴄ}_{3}}\right)+\alpha_{HDPE}\frac{∆T}{K_{T}},$$
where $\alpha_{HDPE}$ is assumed equal to 200 µε/°C and $K_{T}$ is a scale temperature coefficient, empirically determined, for considering the thermal inertia of the pipe.

If all the parameters of Equation (9) can be considered constant along the longitudinal dimension of the pipe, the hoop strain is uniform on this dimension. Otherwise, a local perturbation of 1 or more of them also generates a perturbation of $\varepsilon_{\theta }$ that can be detected from the DFOS helical layout by tuning the cable coil spacing properly according to the parameter that has to be monitored.

During this experimental phase, we used HDPE pipe bars of 2 m length with an external diameter of 63 mm and different Nominal Pressures (PN), i.e., different thicknesses. Specifically, PN10, PN16, and PN25 bars have been tested. Each type of bar has been pressurized with water using a hydrostatic pump connected to 1 of the edges (Figure 4). On the other edge, a vent valve has been installed to remove any internal air bubbles.

In each experiment, the pipe was pressurized to a given value—5 or 10 bar—and then the deforming behavior was monitored over time using DFOS.

### 2.3. Phase II: Detection of Water Leakage

Once positively tested the sensitivity of the sensor layout adopted in the previous phase, we focused on the detection of pressure anomalies along the pipe given, for instance, by water leakage.

The theoretical background is mainly based on the distributed head losses coefficient J. If we consider the usual Darcy–Weisbach equation for an incompressible fluid in a turbulent flow, J is equal to
(10)$$J=\lambda \frac{V^{2}}{2gD}$$
where λ is the friction factor, V is the mean flow velocity, g is the gravity acceleration, and D is the internal pipe diameter. For the estimation of λ, the Colebrook–White equation [41] has been used.

If we consider a length ∆L of a uniform pipeline without concentrated head losses, the total head loss ∆H produced by J is:(11)∆H=JΔL

Pressure drop ∆P is then linked to head loss
(12)∆p=γ∆h=γ∆H,
where g [kg/m^3^] is the specific weight of water and ∆h the piezometric height loss.

Given these considerations, a water leakage generates downstream a lower discharge, hence a lower mean velocity and a lower J along the main pipe. This means that if we take 2 DFOS monitored pipeline sections of the same length ∆L upstream and downstream of the leakage point, we have 2 different ∆P values, hence 2 different hoop strain difference $\Delta \varepsilon_{\theta ,upstream}$ and ${\Delta \varepsilon }_{\theta ,downstream}$. This strain, given the temporal scale of the phenomenon, can be calculated from Equation (9) by considering only the instantaneous elastic contribution and neglecting creep and thermal components:(13)$$\Delta \varepsilon_{\theta }=2\frac{\beta^{2}}{1-\beta^{2}}\frac{∆p}{E_{1}},$$

The term ν also dropped with respect to the previous case because water is not confined at the edge of the pipes.

In order to be detectable, this strain difference has to be greater than the measurement accuracy, that is, 10 µε. Therefore, for each water pipeline system, there is a minimum detectable leaking discharge that depends on the pipeline’s geometrical and mechanical characteristics, on the water velocity and on the cable coil spacing.

In this phase, we used an HDPE PN10 pipe with an external diameter of 32 mm. An 8 m length pipeline forming a “U” shape (Figure 5) has been created on a horizontal plane. A gate valve in the middle has been installed to simulate the water leakage. As in the previous phase, DFOS have been installed on the outer surface of the pipe. The 2 sensing lines before and after the gate valve are joined using an optical fiber connector. The water is moved by a submersible pump placed in a water tank.

The characteristic curve of the pump has been experimentally determined by varying the discharge and checking the pressure head simultaneously with a pressure transducer. The interpolating curve of the experimental points (blue line) is shown in Figure 6, together with the plant characteristic curve (in case of no leakages) analytically determined.

The pump operating point, verified experimentally, is placed at 1.63 L/s which corresponds to a head of 4.07 m.

During the test, when the gate valve is opened, the pump operating point shifts towards the higher discharge and lower head. This affects the value of the distributed head losses and, consequently, the deformation of the pipe. Through DFOS, we monitored these changes for the localization of the pressure anomaly given by the leakage. 

## 3. Results

### 3.1. Phase I

The main results obtained from this experimental phase are illustrated below. Figure 7 shows BFS profiles and raw BGS measured by the BOTDA interrogator on the PN16 pipe at 0, 5 and 10 bar. 

From Figure 7a, it is clear tat the helically wrapped fiber section starts around 4 m and ends at 12.5 m. The fiber section loosely bonded to the pipe for temperature compensation goes from 18 to 20 m. The higher BFS value for the 0 bar curve section corresponding to the fiber wrapped on the pipe is clearly due to pre-strain induced during the installation, while the difference in BFS of the 5 and 10 bar curves with respect to the reference 0 bar one is due to an increasing hoop strain when the pipe pressure is increased. Figure 7b reports the BGS curves measured for each applied pressure in the central zone of the pipe (8 m). As expected, a clear shift of 73 and 200 MHz due to increased hoop strain with respect to the reference 0 bar BGS is evident. Figure 7c shows the BGS curves measured on the loose fiber approximately in the same position of the pipe (19 m). Due to a stable ambient temperature during this experimental phase, there is no shift between the BGS curves. We also point out that the Signal-to-Noise Ratio (SNR) of the BGS measurement is always greater than 40 dB, thus leading to a highly accurate estimation of the BFS values. 

Figure 8 reports the hoop strain history of the PN16 pipe pressurized to 5 bar. The deformation behavior is typically viscoelastic with a full strain recovery after the unloading instant. After the development of instant elastic strain and the primary creep stage, the deformation process enters the secondary creep stage, where the strain rate is constant with time. Temperature variations, as can be seen from the plot, are negligible during the test, and so are the related effects on measurements and on the material.

Thermal effects are instead evident during the pressure test on the PN25. In this case, the loose part of the DFOS sensor layout measured a temperature variation of up to 4 °C. This variation involves thermal deformation that is not recovered in the unloading stage, as can be seen in Figure 9. 

Also, the applied stress has an important influence on the creep process: the primary and the secondary creep stages vary in shape and duration depending on the entity [42]. This has been observed in the test on the PN10 pipe that has the lowest thickness, and so the highest stress applied. Figure 10 compares the strain response with that recorded with the PN16 pipe. The primary creep stage in PN10 shows a higher retardation time, whereas the second stage is governed by a higher strain rate. 

Based on these experimental observations, we calibrated and validated a Burger viscoelastic model using the mechanical scheme reported in Figure 3. The parameters estimated for each type of pipe are synthesized in Table 2. For parameters ν, $E_{1}$ and $\alpha_{HDPE}$ we assumed the values commonly present in literature. The other parameters have been calibrated on the experimental data by minimizing the sum of the square deviation between the measured and the modeled hoop strain. PN16 and PN25 showed to be governed by the same set of values for $E_{2}$, ${ⴄ}_{2}$, and ${ⴄ}_{3}$, whereas for PN10 these parameters are slightly different due to the previous considerations. Figure 11 compares the experimental results with the strain history given by the model for PN10 and PN16 pipes pressurized to 5 bar.

### 3.2. Phase II

Figure 12 shows the theoretical trends along the pipeline of the hydraulic head and the piezometric height before and after the opening of the valve, simulating the leakage. The valve, placed at 4.20 m, was opened at its maximum, providing an outward discharge equal to 34% of the flowing one when the valve is closed.

As can be seen from the plot, the leakage produces a decrease in the piezometric height upstream, given by an increase in the discharge and so of the velocity. This produces a negative hoop strain all along the upstream pipe, whose entity varies along it according to the new and higher $J_{upstream}$ established.

Otherwise, the downstream pipe undergoes a decrease in the flowing discharge that generates a much lower $J_{downstream}$. This causes an increase in the piezometric height hence a positive hoop strain perceptible at a certain distance from the leakage point. 

Figure 13a shows the experimental hoop strain (dot markers) measured by the DFOS compared to the theoretical prediction (represented by a straight solid line) for both the upstream and downstream sections of the pipeline after the opening of the valve. Experimental hoop strain data have been extracted by processing the BFS acquired before and after the leak (Figure 13b) with the method described in Section 2. The pipeline section upstream of the leakage valve is positioned between 9 and 16 m, whereas the downstream one is from 29 m to 36 m. These positions refer to the real length of the fiber cable. The peak at position 20 m is given by the connector. 

As can be seen from the theoretical solution, the downstream pipe is subject to a greater strain variation along the pipe concerning the pipe upstream of the leak point: at the edge of the former, we estimated about +50 με of difference, whereas, at the same point of the latter, this value was set to −12 με. This is caused by the variation of the J given by the water leakage. Table 3 reassumes the expected variation of J from the theoretical background and the measured ones considering the most distant DFOS point available and not affected by the boundary conditions. The percent error is greater in the upstream pipe due to the very low strain variations, close to the accuracy of the DFOS system.

Regarding the DFOS measurements, those collected on the downstream pipe are locally more dispersed from the estimated value. Some DFOS points are more than 20 με far from the estimated values, whereas on the upstream pipe, all measurements points are within the range ±3 με, hence below the system accuracy. This fact is probably caused by the turbulence produced by the leakage that, given the small dimension of the experimental setup, affects the DFOS measurements on the downstream pipe.

## 4. Discussion and Conclusions

The experimental studies we carried out have returned positive feedback on the use of DFOS for the monitoring of HDPE water pipelines.

First of all, the chosen sensor layout guarantees excellent sensitivity in following the strain history of the pipe. Regarding the spatial resolution, this can be simply tuned by varying the spacing of the DFOS cable coils. The strain accuracy is 10 με, which means that if we consider a PN10 HDPE pipe with an external diameter of 63 mm, all pressure variations higher than 0.016 bar produce a measurable hoop strain. 

By measuring the circumferential or hoop strain during the first phase of the experiments, we managed to calibrate and validate the rheological parameters of a viscoelastic model. Moreover, by performing these tests on pipes with different thicknesses, we demonstrated that this solution could potentially be used for monitoring the state of wear of pipelines. Think, for example, of an aged steel water penstock conveying water to a hydropower plant that has reduced its thickness after several decades of service. By wrapping the DFOS cable around the external penstock surface and knowing the original design thickness for each portion of the line, through the hoop strain measurement, it is possible to do a distributed screening of the actual thickness and localize the sections which do not satisfy the serviceability conditions. If we consider, for instance, a steel penstock having an external diameter of 0.72 m, a thickness of 16 mm and an internal pressure of about 35 bar, DFOS are able to detect thickness variations of the order of 2.5% (0.4 mm).

Regarding the second experimental phase, we applied for the first time the DFOS technology to an HDPE pipeline system with running water. The deployed sensing layout is able to detect pressure anomalies along a pipeline given, for instance, by water leakage. The monitoring concept is based on the effects that these events produce on the distributed head losses slope J—hence on the hoop strain—downstream of the leak point. The water flow, by reducing its discharge, hence its velocity, decreases the friction losses with respect to the operational status. The DFOS system layout implemented is able to detect this variation. Despite some noise affecting DFOS measurements given by the characteristics of the experimental setup, the matching with the expected theoretical values is good. 

On the operational level, the strain accuracy of the interrogation unit sets the minimum extension required for detecting properly the variation of J. Figure 14 shows how the minimum required length changes on a PN10 HDPE pipe with an external diameter of 32 mm and a DFOS cable spacing layout of 4 cm by considering different flowing velocities and different percentages of leakage.

Given the operational flow rate of the pipeline system, the lower the minimum outward discharge that has to be detected, the higher the minimum required DFOS sensorized length. This parameter also depends on the geometrical and mechanical properties of the pipe section.

An important outcome is that by monitoring only a given portion of a pipeline, it is possible to have enough information about the integrity of the upstream part of the system. Hence, a DFOS-based monitoring system is potentially able to cover even several kilometers of pipeline infrastructure by simply installing sensorized pipe elements with a customized spatial frequency. 

In this work, we investigated for the first time the use of Brillouin DFOS for the monitoring of water HDPE pipeline infrastructures, whose management is an increasing issue all over the world. The leak detection principle is based on the monitoring of pipeline hoop strain. This is an indirect measurement of the internal pressure variation caused by a water leak. Therefore, once known the mechanics of the material and the hydraulic characteristics of the water flow, the data processing effort is quite fast. This is an alternative approach with respect to that based on acoustic/vibrational methods, mainly used on steel pipelines. Other strain monitoring solutions, like Fiber Bragg Gratings or vibrating wire extensometers, can provide better accuracy than Brillouin-based DFOS, but they offer only a punctual strain measurement. Hence, their extensive use in a pipeline, as we have done with DFOS, would imply much higher costs and a more complex system layout. 

We proved that this technology is able to continuously detect time and space pressure anomalies affecting the pipeline by changing its hoop strain. This means that we can consider that this technology has achieved an RTL maturity of level 4, “Technology validated in the lab”.

This type of pipeline monitoring goes beyond traditional pipeline pressure monitoring solutions, limited to a given number of characteristic points, and possibly offer, once further evidence has been collected in relevant and operational environments, and some technological industrialization barriers have been overcome, a continuous and simultaneous view of the trend of loads on the entire network. The tests have been performed on HDPE pipes, commonly used for water-conveying infrastructures for agricultural, civil and industrial purposes. The DFOS cable has been wrapped and fixed on the external surface of the pipe. The sensibility of this sensor layout has been positively assessed through our experimental tests performed in the first phase. 

In the second phase, we instead focused on the possibility of detecting the effects related to a water leakage along a pipeline. In particular, we based our investigation on the variation of the distributed head loss coefficient J, having clearly in mind that this is the main parameter that rules pressure losses in long pipeline networks. We focused not only on detecting leakages but also on estimating their entity. Water leaks of the order of 5% are potentially detectable in pipelines with a flowing velocity between 2 and 4 m/s by simply tuning the length sensorized with DFOS. The theoretical and experimental outcomes are promising, and the operational characteristics of this technology, again to be considered as achieving an RTL level of 4, are compatible with on-field applications. The energy supply is required only by the interrogation device unit that can be placed even several kilometers from the monitored section of the pipeline network. 

Future developments will be focused on more detailed studies of this monitoring solution (real environment included) and on the industrial production of HDPE pipes “natively smart,” in which DFOS cables are integrated into the pipeline surface during the extrusion process. The compensation of thermal effects produced by thermal gradients present along the pipeline path in the soil trench is also another challenge that could be potentially solved by installing a loose DFOS cable beside the pipeline. Finally, a DFOS data interpretation layer based on machine learning algorithms could be developed to create a leak prediction model.

## Figures and Tables

**Figure 1 sensors-23-06205-f001:**
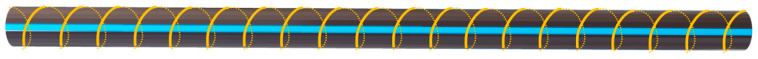
A scheme of the sensor layout adopted: the DFOS cable is helically wrapped on the external surface of the pipe.

**Figure 2 sensors-23-06205-f002:**
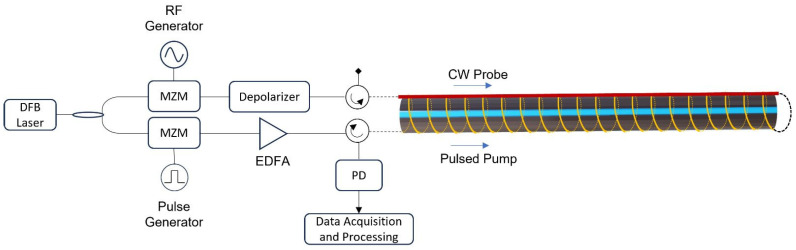
Block diagram of the BOTDA unit. In yellow is the helically wrapped optical fiber bonded to the HDPE pipe for strain sensing, and in red is the returned fiber, loosely fixed to the pipe, for temperature compensation.

**Figure 3 sensors-23-06205-f003:**
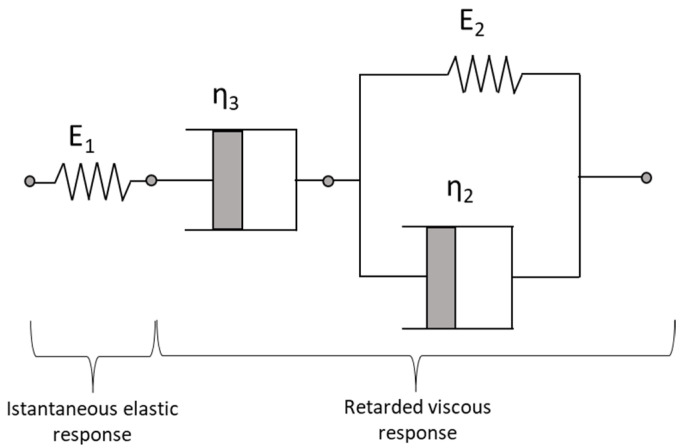
Burger model consisted of Maxwell and Kelvin–Voigt models in series.

**Figure 4 sensors-23-06205-f004:**
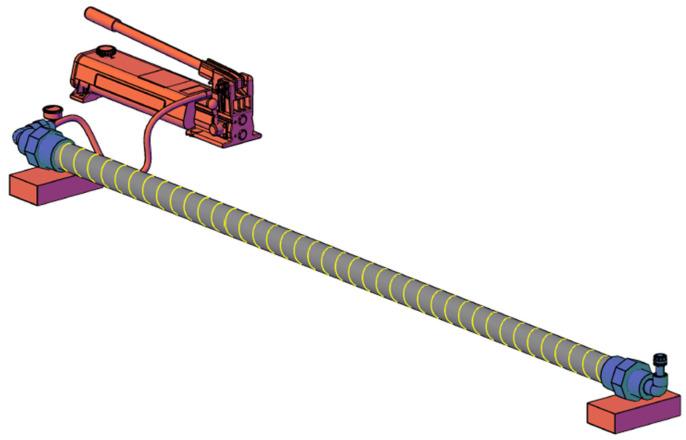
Phase I experimental setup. The bar is pressurized using a hand test pump. On the other edge, a vent valve prevents the formation of air bubbles.

**Figure 5 sensors-23-06205-f005:**
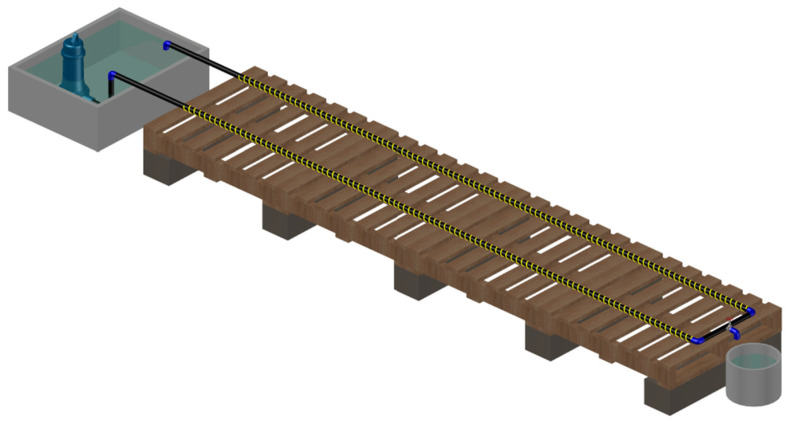
Phase II experimental setup. An 8 m length HDPE pipeline has been created on a horizontal plane with a gate valve to simulate a water leakage. The water is moved by a submersible pump connected to one edge of the pipeline.

**Figure 6 sensors-23-06205-f006:**
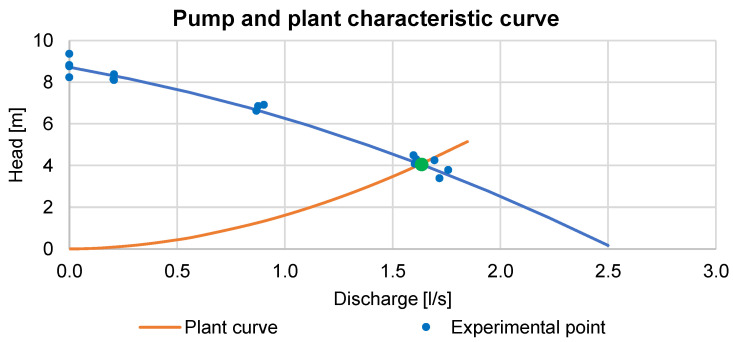
Characteristic pump and plant curve.

**Figure 7 sensors-23-06205-f007:**
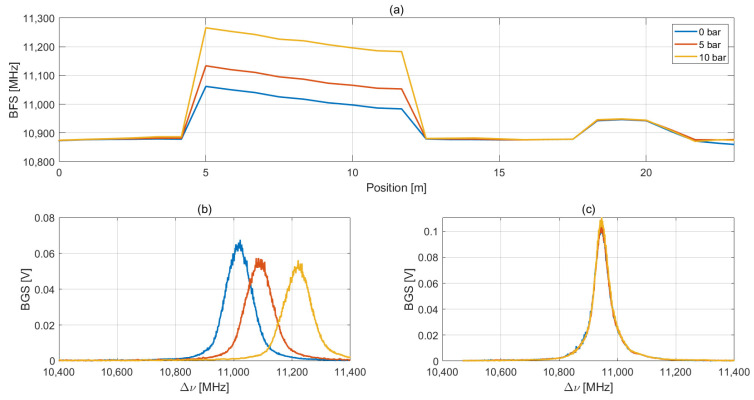
(**a**) Measured BFS profiles for different pressure values applied to the pipe. (**b**) BGS profiles measured on the helically wrapped sensing fiber in the central part of the pipe (8 m). (**c**) BGS measured on the temperature-compensation fiber in the same position as (**b**).

**Figure 8 sensors-23-06205-f008:**
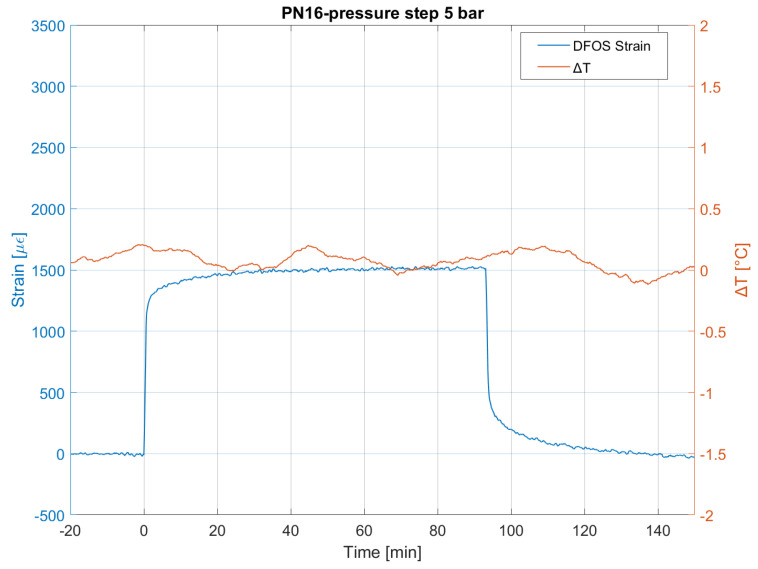
Mean spatial strain and temperature variation measured by DFOS during the pressure test on the PN16 pipe.

**Figure 9 sensors-23-06205-f009:**
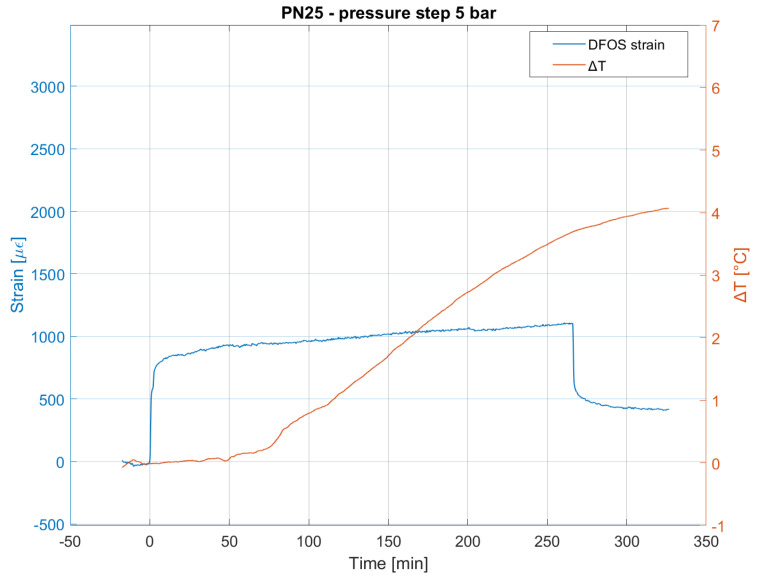
Mean spatial strain and temperature variation measured by DFOS during the pressure test on the PN25 pipe.

**Figure 10 sensors-23-06205-f010:**
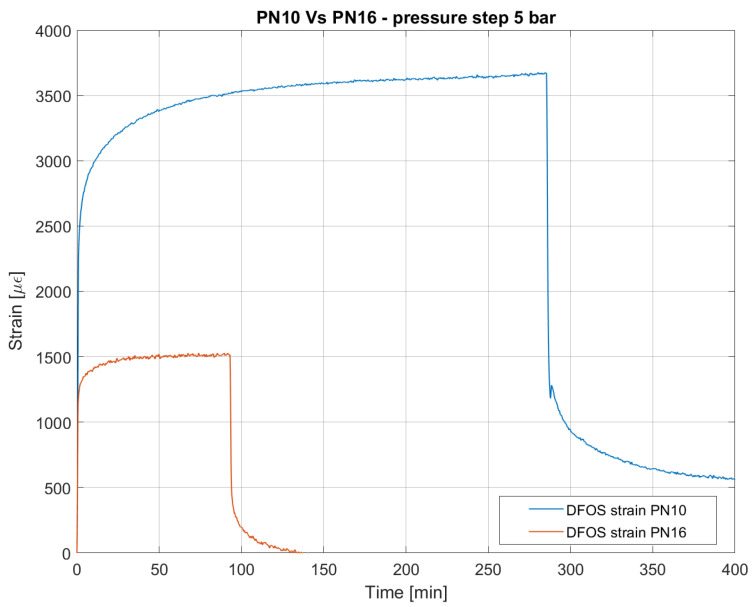
Comparison between mean spatial strain measured by DFOS on PN10 and PN16 pipes pressurized to 5 bar. Both experiments have been performed at a constant temperature.

**Figure 11 sensors-23-06205-f011:**
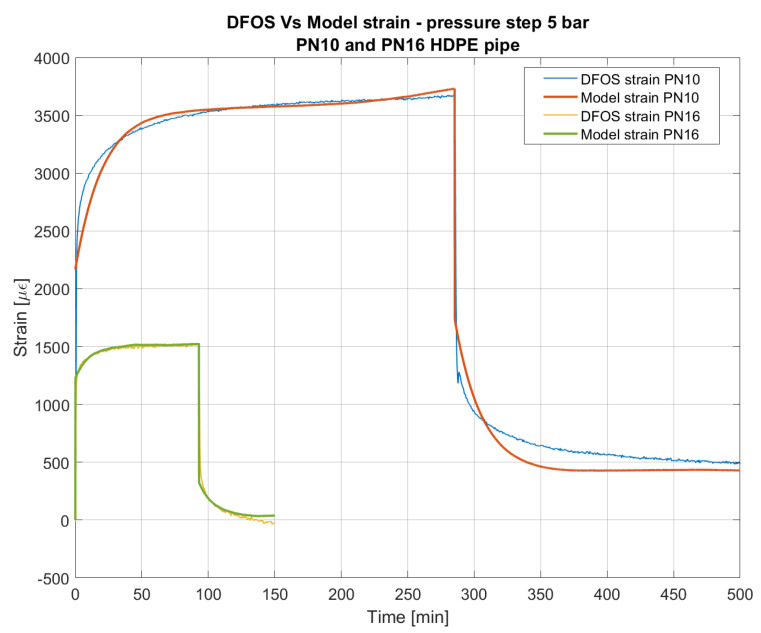
Comparison between the strain measured by DFOS and the strain modeled using the Burger creep model for PN10 and PN16 HDPE pipe pressurized to 5 bar.

**Figure 12 sensors-23-06205-f012:**
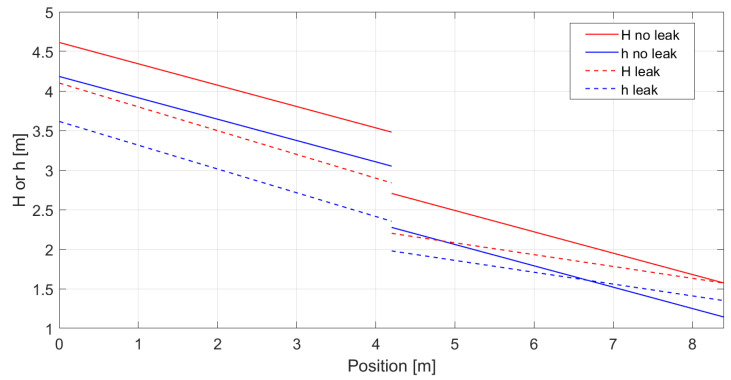
Hydraulic head and piezometric height along the pipeline before (full lines) and after (dashed lines) the opening of the leakage valve.

**Figure 13 sensors-23-06205-f013:**
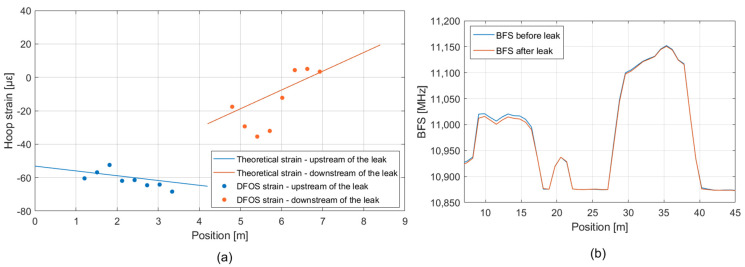
(**a**) Comparison between the theoretical hoop strain and the DFOS measurements along the pipeline after the opening of the leakage valve. (**b**) BFS profiles along the pipeline measured before and after the opening of the leakage valve.

**Figure 14 sensors-23-06205-f014:**
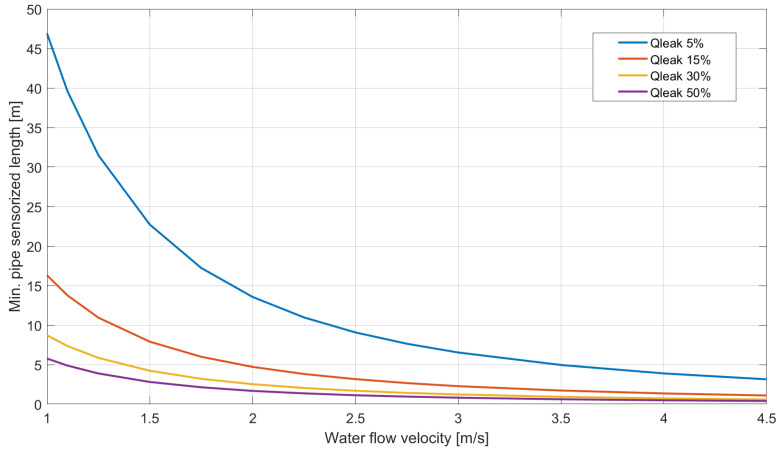
Minimum pipe sensorized length required to detect a change on *J* by considering different flowing velocities and different percentages of leaking discharge with respect to the operational status.

**Table 1 sensors-23-06205-t001:** DFOS coil length, DFOS coil spacing, and effective strain and temperature spatial resolution for each experimental phase.

Phase	$L_{C}$ [m]	S [m]	Δs [m]	Δν [GHz]	$\nu_{s}$ [MHz]	$N_{avg}$	∆t [s]
I	0.198	0.05	0.45	10.4–11.4	5	256	7
II	0.108	0.04	0.72	10.75–11.3	1	256	20

**Table 2 sensors-23-06205-t002:** Geometrical and mechanical parameters were estimated for each type of pipe tested.

Parameter	u.m.	PN10	PN16	PN25
*β*	-	0.86	0.78	0.70
*ν*	-	0.4	0.4	0.4
$E_{1}$	MPa	1.05 × 10^3^	1.05 × 10^3^	1.05 × 10^3^
$E_{2}$ ${ⴄ}_{2}$	MPa	1.65 × 10^3^	4.47 × 10^3^	4.47 × 10^3^
	MPa s	2.04 × 10^6^	2.77 × 10^6^	2.77 × 10^6^
${ⴄ}_{3}$	MPa s	2.13 × 10^8^	4.68 × 10^8^	4.68 × 10^8^
$\alpha_{HDPE}$	µε/°C	200	200	200
$K_{T}$	-	4	4	4

**Table 3 sensors-23-06205-t003:** Expected and measured variation of *J* upstream and downstream of the leakage point.

Pipe	Δ*J* Estimated	Δ*J* Measured	Error %
Upstream of leak	−0.03	−0.04	30.27
Downstream of leak	+0.12	+0.11	12.31

## Data Availability

The datasets are available upon request to the corresponding author.

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
