# Peer review of "Experimental Investigations of Distributed Fiber Optic Sensors for Water Pipeline Monitoring"

_sensors, 2023, doi:10.3390/s23136205_

Round 1

Reviewer 1 Report

In this paper, water loss is monitored in water pipeline networks using BOTDA system. The authors have verified the ability to detect strain related to pressure anomalies and caused by water leakage along a pipeline. Static and dynamic pressure on the water pipeline is also experimentally measured by the proposed sensing system. In my point of view, the manuscript can be accepted after addressing the following concerns:

1.       The authors used the sensing cable at the end of the pipe for temperature compensation, could the authors comment on the accuracy of this method if the monitored pipeline is long, in which the temperature distribution may vary along the pipeline? Any other practical solution to this issue?

2.       Helically wrapping the sensing cable around the pipeline may inevitably introduce pre-strain on the fiber before strain or pressure caused by water leak is induced on the fiber and this pre-strain (or bending/torsion pressure) is different to be maintained along the entire sensing cable, would this place an impact on the strain measurement along the pipeline? The authors should justify the advantages of using helically wrapping method.

3.       The measure distributed strain for downstream of the leak somehow shows an approximate sinusoidal shape, any explanations on this phenomenon?

4.       “Errore. L'origine riferimento non è stata trovata.,” appears twice in Line 286 and Line 298, what does this mean?

Minor editing of English language required

Author Response

Dear Reviewer,

The authors want to thank you for the time and attention dedicated to our work. We received helpful suggestions that certainly will enrich the contents of the article. Here attached our reply to your comments and suggestions. Changes in the manuscript have been implemented accordingly. 

Kind regards,

Reviewer 2 Report

The problem of high-resolution pipeline monitoring is of great practical interest. The topic of this publication is relevant. The article is well structured and presented. I believe this article can be published with minimal edits. It would be desirable to have more details on the experimentally tested pipeline sensor resolution. It would also be useful to compare the sensitivity and resolution of the sensor shown in the article with other sensors for pipeline monitoring. Such numerical evaluations and comparisons would make it easier for readers to realize the advantages and weaknesses of the approach. In particular, the authors in the introduction excluded the use and competitiveness of acoustic sensor approaches without sufficient reasoning (lines 67-68).

In the article authors presented the experimental results of using distributed fiber optic sensors based on stimulated Brillouin scattering technology to detect strain related to pressure anomalies along a pipeline caused by water leakage.
The problem of monitoring pipelines is of great practical interest, so the topic of the publication is relevant.
The article is well written, well structured and presented.
I believe that this article corresponds to the theme of the "Sensors" journal and can be published in it after a little revision.
The introduction can be extended by comparing different physical pipeline monitoring methods in order to better justify the chosen approach. In particular, the authors in the introduction excluded the use and competitiveness of acoustic sensor approaches without sufficient reasoning (lines 67-68). I cannot agree with such a categorical statement and recommend the authors to revise this part using respective Refs.
The article does not sufficiently indicate the parameters of the measuring equipment, they could be added briefly in the section on materials and methods.
It would be desirable to have more details on the experimentally tested pipeline sensor resolution. It would also be useful to compare the sensitivity and resolution of the sensor shown in the article with other sensors for pipeline monitoring (with adding relevant bibliographic references). What is the signal to noise ratio? How important is post-processing of the experimental data in this technique? Some numerical evaluations and comparisons would make it easier for readers to realize the advantages and weaknesses of the used approach.
In addition, I would recommend that there be more discussion about the accuracy of the leak detection technique and the factors that can affect it.
The work looks too applied, and to increase its scientific value, I would recommend adding more analysis of physical phenomena in the discussion section. In addition, it is desirable for the authors to more clearly define what is the scientific novelty and main achievements of their work (as far as I can see, the technique is rather standard).

Author Response

Dear Reviewer,

thank you for your observations and suggestions. We have revised and integrated the manuscript according to your review. Here attached the document containing our reply to each of your points.

Kind regards, 

Reviewer 3 Report

The authors presented a rather interesting work on distributed monitoring using BOTDA. They have worked on High Density Polyethylene (HDPE) pipes and winded and fixed the optic fiber cable on the pipe external surface, and then verified the ability to detect strain related to pressure anomalies along a pipeline, e.g. those caused by water leakage. It seems to me that the paper contains attractive practical knowledge and will be of interest to readers of Sensors. However, I have a few notes that need to be discussed before publishing:

1. Unfortunately, the hyperlinks refering to the figures do not work in the manuscript, and instead of them an error message appears in Italian. I ask the authors to fix this;

2. When choosing a research technology, the authors justify the rejection of the DAS method by the high cost of such instruments. State-of-the-art data processing methods allow one the use of cheaper components with a total cost of up to 20,000 USD and achieve impressive signal-to-noise ratios [1,2], and low-frequency detection methods [3,4,5] will certainly be able to capture larger leaks. Is BOTDA technology, which still requires access to both line inputs, much cheaper?;

3. For a complete understanding and repeatability of the experiment, as well as for a clearer idea of how the signal is formed and registered, one of the two must be provided:
- Scheme of the BOTDA setup created in the laboratory, showing the units/components and their models;
- An accurate model/id of a commercial BOTDA used in the experiment containing the main characteristics: dynamic range, typical pulse durations, number of averages, etc. This would be great if you arrange this as a table;

4. For a clearer understanding of the experimental curve behavior in Figure 11, I would ask the authors to add the original trace or color surface obtained using BOTDA, as well as one of the original BGS, (Brillouin gain spectrum). Since the detailed characteristics of the setup are not given, despite the fact that the line under test is rather short, and deformations of 40 microstrains do not lead to a serious increase in attenuation in the fiber, one should make sure that the spectrum has a sufficient signal-to-noise ratio to obtain data without the use of special the Brillouin frequency shift (BFS) extraction methods.

5. Is the BFS itself significant for the final result? What I mean is that the authors want microstrains, but not frequency shift, as their desired final data. Practice shows that on a particular object it is sometimes easier to use a neural network, the input of which was the whole BGS [5,6]. As also known, this technique is also applicable for detecting leaks in pipelines [7]. Perhaps the authors may consider this as a future work.

6. Please specify the thight buffer material.

7. Some figures are located right at the end of the article sections. I would suggest moving them higher, right after the first mention in the text.

[1] http://dx.doi.org/10.3390/s22239482
[2] http://dx.doi.org/10.3390/photonics10040463
[3] https://doi.org/10.3390/s23125600
[4] https://doi.org/10.1016/j.yofte.2020.102298
[5] http://dx.doi.org/10.3390/s22072677
[6] https://doi.org/10.3390/s23063226

Author Response

(The authors gave the same response as above.)
